# Clinical presentation and comorbidities of obstructive sleep apnea-COPD overlap syndrome

**Dan Adler** [1,2] *, **Sébastien Bailly**[3,4], **Meriem Benmerad**[3,4], **Marie Joyeux-Faure**[3,4], **Ingrid Jullian-Desayes**[3,4], **Paola Marina Soccal**[1,2], **Jean Paul Janssens**[1,2], **Marc Sapène**[5], **Yves Grillet**[6], **Bruno Stach**[7], **Renaud Tamisier**[3,4], **Jean-Louis Pépin**[1,3,4]

1 Division of Pneumology, Geneva University Hospitals, Geneva, Switzerland, 2 University of Geneva Faculty of Medicine, Geneva, Switzerland, 3 HP2 Laboratory, INSERM U1042, University Grenoble Alpes, Grenoble, France, 4 EFCR Laboratory, Pole Thorax et Vaisseaux, Grenoble Alpes University Hospital, Grenoble, France, 5 Private Practice Sleep and Respiratory Disease Centre, Nouvelle Clinique Bel Air, Bordeaux, France, 6 Private Practice Sleep and Respiratory Disease Centre, Valence, France, 7 Private Practice Sleep and Respiratory Disease Centre, Valenciennes, France

* dan.adler@hcuge.ch

**Data Availability Statement:** All relevant data are within the paper and its Supporting Information files.

## Abstract

### Background

More advanced knowledge is needed on how COPD alters the clinical presentation of obstructive sleep apnea (OSA) and how the association of both diseases, known as 'overlap syndrome' (OVS), impacts on cardiovascular health.

### Objective

To investigate differences between patients with OVS and those with moderate-to-severe OSA alone.

### Methods

A cross-sectional study conducted in the French National Sleep Apnea Registry between January 1997 and January 2017. Univariable and multivariable logistic regression models were used to compare OVS versus OSA alone on symptoms and cardiovascular health.

### Results

46,786 patients had moderate-to-severe OSA. Valid spirometry was available for 16,466 patients: 14,368 (87%) had moderate-to-severe OSA alone and 2098 (13%) had OVS. A lower proportion of OVS patients complained of snoring, morning headaches and excessive daytime sleepiness compared to OSA alone (median Epworth Sleepiness Scale score: 9 [interquartile range (IQR) 6–13] versus 10 (IQR 6–13), respectively; $P$ <0.02). Similarly, a lower proportion of OVS patients (35.6% versus 39.4%, respectively; $P$ <0.01) experienced sleepiness while driving. In contrast, 63.5% of the OVS population experienced nocturia compared to 58.0% of the OSA population ($P$<0.01). Apnea hypopnea index (36 [25; 52] vs 33.1 [23.3; 50]), oxygen desaturation index (28 [15; 48] vs 25.2 [14; 45]) and mean nocturnal

**Funding:** This study was funded by an unrestricted grant from "Appel d'offre cohorte-fondation du souffle" (SRC_2017) to JLP and RT. RT and JLP are supported by a research grant from the French National Research Agency (ANR-12-TECS-0010) in the framework of the "Investissements d'avenir" programme (ANR-15-IDEX-02) and the "e-health and integrated care" Chair of excellence of the University Grenoble Alpes Foundation. DA is supported by research grant LPS-16/12 from the "Ligue Pulmonaire Suisse" to conduct a programme entitled "integrated care of patients surviving acute hypercapnic respiratory failure in the ICU."

**Competing interests:** The authors have declared that no competing interests exist.

SaO2 (92 [90; 93.8] vs 93 [91.3; 94]) were significantly more altered in the OVS group. Associated COPD had no effect on the prevalence of hypertension and stroke. After controlling for main confounders, COPD severity was associated in a dose-response relationship with a higher prevalence of coronary heart disease, heart failure and peripheral arteriopathy.

## Conclusions

In adults with moderate-to-severe OSA, OVS was minimally symptomatic, but exhibited higher odds for prevalent coronary heart disease, heart failure and peripheral arteriopathy.

## Introduction

Obstructive sleep apnea (OSA) and COPD are among the most prevalent chronic diseases and represent a major economic burden for healthcare systems worldwide [1–3]. Two recent statements from the American Thoracic Society and the European Respiratory Society have emphasized that the clinical presentation of OSA-COPD "overlap syndrome" compared to OSA alone remains undefined and more research is needed to elucidate how the association of both diseases impacts on cardiovascular health [4, 5]. In particular, observational studies comparing cardiovascular outcomes among individuals with OVS compared to OSA or COPD alone have been recommended to address this knowledge gap [4].

OSA can be suspected in various clinical settings, including in highly symptomatic individuals or in an 'at-risk' population for cardiometabolic diseases where most patients do not present classical 0SA symptoms. This has now been evidenced by multiple recent cluster analyses [6–9]. The description of additional phenotypes is ongoing and an improved knowledge of their associated clinical presentation is required to appropriately design diagnostic pathways and specific patient-reported outcomes adapted to various clinical scenarios. In line with this challenge, we and others have reported that OSA is less symptomatic in COPD patients and might be better anticipated by assessment of lung function rather than classical screening tools [10–12]. However, these data emerge from relatively limited sample size populations and need external validation in large real-life routine practices to allow a generalization of findings.

The co-occurrence of OSA and COPD is characterized by profound nocturnal hypoxemia and a higher risk of pulmonary hypertension and right heart failure [13–16]. In terms of intermediary mechanisms, OSA and COPD share an increased burden of systemic inflammation via the activation of transcriptional factors (nuclear factor kappa B and hypoxia-inducible factor 1), oxidative stress, sympathetic overactivity and endothelial dysfunction [17–19]. Thus, it can be expected that cardiovascular diseases are more prevalent in patients with OVS compared to OSA or COPD alone. However, there are very few reports on the prevalence of cardiovascular diseases in patients with OVS [20–23] and it remains to be delineated how OSA and COPD interact to magnify the cardiometabolic risk in real-life routine practice.

The aim of this study was to report clinical differences between OSA and OVS patients at initial presentation in a large national sleep apnea registry and to investigate the effect of COPD on the cardiovascular risk in OSA patients.

## Methods

### Study design and population

This is a cross-sectional study using prospective data from the French National Sleep Apnea Registry (Observatoire Sommeil de la Fédération de Pneumologie [OSFP; www.osfp.fr]). The

methodology and participants of the registry have been described in detail elsewhere [9, 24, 25]. In brief, the registry is a web-based report administered by the French Federation of Pulmonology. It contains standardized, anonymized, longitudinal data of patients referred for suspected sleep-disordered breathing, assessed by respiratory physicians working in private practice and general or university hospitals.

Among 100,759 patients included in the OSFP database from January 1997 to January 2017, 72,376 patients had completed a sleep study within 6 months of the first medical visit and were flagged for analysis in the present study. A subgroup of 46,786 patients >18 years old with an apnea-hypopnea index (AHI) ≥15 and/or oxygen desaturation index (ODI) ≥15 was then selected to assure a robust diagnosis of sleep apnea. Valid measurement of forced expiratory volume in 1 second ($FEV_1$) over forced vital capacity (FVC) was available for 16,466 patients (Fig 1). Demographic data and medical history were collected during the baseline evaluations, including diagnosis of cardiovascular comorbidities made by a physician. Symptoms were evaluated using standardized questionnaires.

Ethics committee approval to set up the study database was obtained from the French information technology and personal data protection authorities (C.C.T. I.R.S n° 09.521). The OSFP Independent Scientific Advisory Committee approved the use of this database for the present study. All patients included gave written informed consent.

## Sleep studies

Full night, in-laboratory, attended polysomnography or type III cardiopulmonary sleep recordings are required by the OSFP registry for OSA diagnosis. In France, sleep stages and arousals are usually scored using the American Academy of Sleep Medicine criteria [26]. Apnea is scored if a drop of 90% or more in airflow signal excursion is noted for at least 10 s. Hypopnea is defined as a drop greater than or equal to 30% in airflow lasting at least 10 s and associated with 3% oxygen desaturation or electroencephalogram arousal. Type III cardiopulmonary sleep recordings are a reliable, cost-effective option for suspected OSA [27, 28]. Interpretation of type III recording using hypopnea criteria, which includes only a 30% drop in airflow and 3% desaturation, has been demonstrated to have the best diagnosis accuracy compared to full polysomnography [29]. AHI cut-off values between 15 and <30 are used to classify moderate sleep apnea and a cut-off value ≥30 is considered as severe sleep apnea.

## Spirometry

In France, pulmonary function tests are offered as part of the initial workup for OSA in the case of current or past cigarette smoking and/or obesity (body mass index [BMI] >30 kg/m2) and/or respiratory symptoms, such as dyspnea [30]. A post-bronchodilator fixed $FEV_1$/FVC ratio <70% is used to define the presence of persistent airflow limitation and thus COPD.(1) Specific spirometric $FEV_1$ (percentage of predicted) cut points are used to assess airflow limitation severity: 1) GOLD 1 ($FEV_1$ ≥80%); 2) GOLD 2 (50% ≤$FEV_1$ <80%); 3) GOLD 3 (30% ≤$FEV_1$ <50%); 4) GOLD 4 ($FEV_1$ <30%).

## Statistical analysis

Descriptive statistics are reported as counts and percentages for categorical data and the median and interquartile range (IQR) for continuous variables. When comparing subgroups, we used the Mann-Whitney-Wilcoxon test for continuous variables or the chi-squared test for categorical variables. As variables presented less than 5% of missing values, a simple imputation was implemented, based on the median for quantitative variables or most common attribute value for qualitative data. Univariable logistic regression models were used to characterize

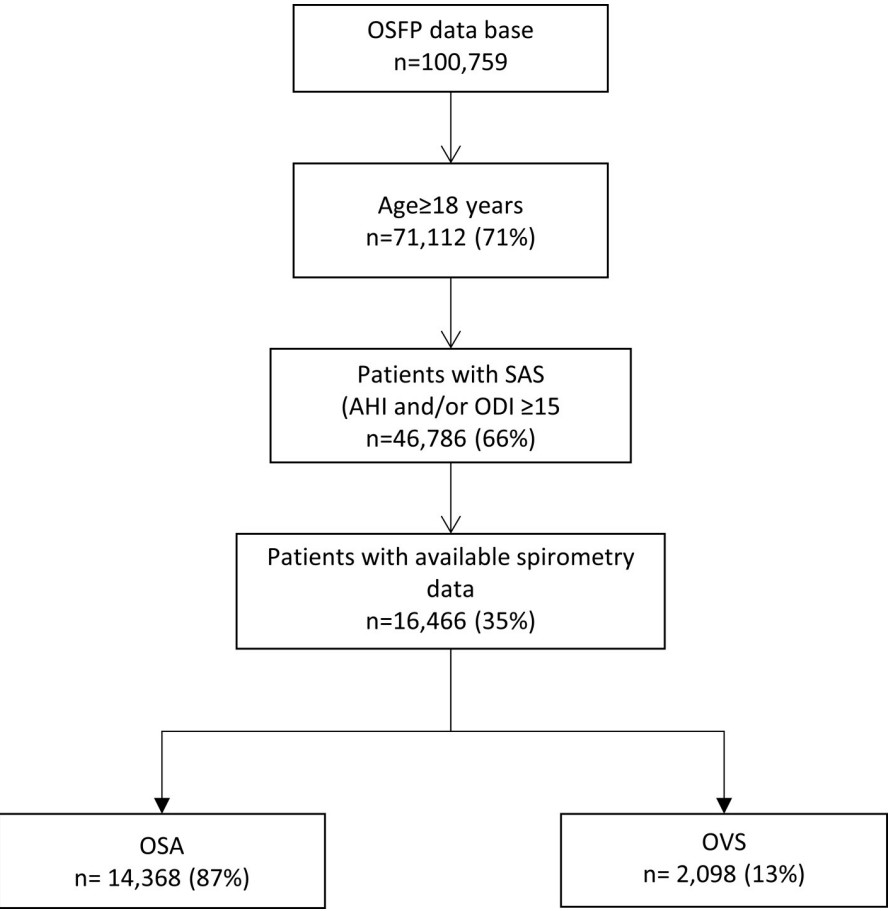

**Fig 1. Study population profile.** OSFP = Observatoire Sommeil de la Fédération de Pneumologie, OSA = sleep apnea syndrome, AHI = apnea-hypopnea index, ODI = oxygen desaturation index, OVS = overlap syndrome.

OVS compared to OSA alone. Age, gender, apnea-hypopnea index, hyperlipidemia, hypertension, peripheral arterial disease, heart failure, coronary artery disease and myocardial infarction, stroke, diabetes, cigarette smoking and alcohol exposure, excessive daytime sleepiness and nocturia were all significantly associated with the outcome ($P < .05$) and included in a multivariable logistic regression model. A stepwise forward selection method was used to define the final multivariable model. Statistical interaction between $FEV_1$ and AHI on cardiovascular outcomes was non-significant when tested in a logistic model ($P$ values of $>0.15$ for all cardiovascular outcomes). Logistic regression models were also used to test the effect of type of sleep study (type III or type I), age, gender, BMI, smoking status, AHI and COPD on cardiovascular outcomes. The collinearity between factors was tested. Log linearity between the variables was tested and the variables were transformed into categorical variables if necessary. Statistical analyses were conducted with SAS v9.4 (SAS Institute Inc., Cary, NC). Results were considered significant at $P$ values of 0.05.

## Results

### Study population

A total of 16,466 patients with available spirometry data were included in the present analysis. The clinical characteristics are presented in Table 1. Among these, 14,368 (87%) had OSA and

**Table 1. Study population.**

| Variable | Items | ALL n = 16,466 | OSA n = 14,368 | Overlap n = 2098 | P | Missing (%) |
|---|---|---|---|---|---|---|
| Age, yrs, [IQR] | | 57.7 [48.6;66.1] | 56.9 [47.7; 65.1] | 63.5 [55.7; 71.7] | < .01 | |
| Male gender (%) | | 11,693 (71.2) | 10,004 (69.8) | 1689 (80.7) | < .01 | 38 (0.2) |
| Body mass index, kg/m$^2$,median [IQR] | | 31.1 [27.3; 35.9] | 31.1 [27.3; 35.9] | 31.1 [27.3; 35.9] | 0.72 | 186 (1.1) |
| Smoking status | Non-smoker | 8,019 (48.8) | 7,316 (51.0) | 703 (33.7) | < .01 | 48 (0.3) |
| | Ex-smoker | 5,434 (33.1) | 4,495 (31.4) | 939 (45.0) | . | |
| | Smoker | 2,965 (18.1) | 2,522 (17.6) | 443 (21.2) | . | |
| Alcohol use, n (%) | | 816 (5.0) | 623 (4.4) | 193 (9.3) | < .01 | 87 (0.5) |
| Sedentarity, n (%) | | 3,525 (21.5) | 3,014 (21.1) | 511 (24.5) | < .01 | 81 (0.5) |
| FEV$_1$/FVC, [IQR] | | 81 [75; 87] | 82 [77; 87.9] | 64 [59; 67] | < .01 | |
| FEV$_1$ (% predicted), [IQR] | | 95 [83; 106] | 97 [86; 107.7] | 75 [60; 88.1] | < .01 | |
| COPD status, n (%) | No | 14,368 (87.3) | 14,368 (100) | 0 | / | |
| | GOLD Stage 1 | 863 (5.2) | 0 | 863 (41.1) | | |
| | GOLD Stage 2 | 976 (5.9) | 0 | 976 (46.5) | | |
| | GOLD Stages 3 & 4 | 259 (1.6) | 0 | 259 (12.3) | | |
| Apnea-hypopnea index, [IQR] | | 34 [24; 50.6] | 33.1 [23.3; 50] | 36 [25; 52] | < .01 | 16 (0.1) |
| Oxygen desaturation index, [IQR] | | 26 [14.; 45.2] | 25.2 [14; 45] | 28 [15; 48] | < .01 | 2988 (18.1) |
| Mean nocturnal Sa02, [IQR] | | 93 [91; 94] | 93 [91.3; 94] | 92 [90; 93.8] | < .01 | 3571 (21.7) |
| Sleep examination, n (%) | Polysomnography | 6,200 (39.8) | 5,323 (39.1) | 877 (44.1) | < .01 | 870 (5.3) |
| | Polygraphy | 9,396 (60.2) | 8,285 (60.9) | 1,111 (55.9) | . | |
| pH, [IQR] | | 7.4 [7.4; 7.5] | 7.4 [7.4; 7.5] | 7.4 [7.4; 7.5] | 0.06 | 10,256 (62.3) |
| PaCO2, mmHg [IQR] | | 38 [35.8; 41] | 38 [35.6; 41] | 39 [36; 43] | < .01 | 8748 (53.1) |
| PaCO2≥ 45 mmHg, n(%) | | 649 (8.4) | 445 (6.7) | 204 (18.6) | < .01 | 8748 (53.1) |
| PaO2, mmHg [IQR] | | 79 [71; 85.8] | 79 [72; 86] | 74.1 [67; 81.1] | < .01 | 8726 (53) |

OSA = obstructive sleep apnea, FEV$_1$ = forced expiratory volume in 1 second, FVC = forced vital capacity.

2098 (13%) had OVS. Clinical characteristics of 16,466 OSA patients who were offered spirometry compared to 30,320 OSA patients without available spirometry data are available in the online supplementary S1 Table. By definition, OVS patients had a median FEV$_1$/FVC ratio of 64% (interquartile range [IQR] 59 to 67) compared to OSA patients who had a FEV$_1$/FVC ratio of 82% (IQR 77 to 87.9) (P <0.01). Median FEV$_1$ (% predicted) was 75 (IQR 60.0 to 88.1) in OVS patients and 97 (IQR 86.0 to 107.7) in OSA patients (P <0.01). GOLD stages were: 1: 863 (41%); 2: 976 (46.5%); 3 and 4 combined: 259 (12.3%). There were no differences in BMI between groups. Mean nocturnal SaO2 was lower in patients with OVS versus OSA alone (92% (IQR 90.0 to 93.8) and 93% (IQR 91.3 to 94.0), respectively; P <0.01). Conversely, PaCO$_2$ was higher in patients with OVS compared to patients with OSA (39 mmHg (IQR 36.0 to 43.0) versus 38 mmHg (IQR 35.6 to 41.0), respectively; P <0.01). OVS exhibited a higher percentage of chronic hypercapnia (PaCO2> 45 mmHg) (18.6% (95% CI 16.3 to 20.0) versus 6.7% (95% CI 6.11 to 7.32), respectively; P <0.01).

## Anthropometric factors and exposures associated with OVS

Older age, male gender, higher exposure to cigarette smoke and alcohol and reduced reported physical activity were associated with OVS. After adjustment for confounders, the same clinical factors remained associated with OVS in the multivariable analysis, except for reduced physical activity. Table 2 shows the results of the univariable and multivariable logistic regression models depicting factors associated with OVS.

**Table 2. Predictors of OVS in the study population.**

| Variable | | Univariable analysis OR [95% CI] | P | Multivariable analysis OR [95% CI] | P |
|---|---|---|---|---|---|
| Age (10 years) | | 1.56 [1.50; 1.63] | <0.01 | 1.64 [1.58; 1.72] | <0.01 |
| Gender (men) | | 1.80 [1.61; 2.02] | <0.01 | 1.46 [1.29; 1.65] | <0.01 |
| Smoking status | Non-smoker | 1 | <0.01 | 1 | <0.01 |
| | Ex-smoker | 2.14 [1.93; 2.38] | | 1.83 [1.64; 2.04] | |
| | Smoker | 1.80 [1.59; 2.05] | | 2.45 [2.14; 2.82] | |
| Peripheral arteriopathy | | 3.31 [2.66; 4.11] | <0.01 | 1.88 [1.50; 2.35] | <0.01 |
| Alcohol | | 2.24 [1.89; 2.65] | <0.01 | 1.69 [1.4; 2.02] | <0.01 |
| Apnea-hypopnea index (event/hour) | <24 | 1 | <0.01 | | |
| | [24; 34] | 1.02 [0.89; 1.17] | | | |
| | [34; 50.6] | 1.26 [1.10; 1.43] | | | |
| | ≥50.6 | 1.24 [1.09; 1.41] | | | |
| Sedentarity | | 1.21 [1.09; 1.35] | <0.01 | | |
| Hyperlipidemia | | 1.25 [1.14; 1.38] | <0.01 | | |
| Hypertension | | 1.46 [1.33; 1.60] | <0.01 | | |
| Coronary artery disease and myocardial infarction | | 1.91 [1.66; 2.19] | <0.01 | | |
| Heart failure | | 2.06 [1.63; 2.62] | <0.01 | | |
| Stroke | | 1.53 [1.21; 1.93] | <0.01 | | |
| Diabetes | | 1.47 [1.31; 1.66] | <0.01 | | |
| Sleepiness at the wheel | | 0.85 [0.77; 0.93] | <0.01 | | |
| Nocturia | | 1.26 [1.15; 1.39] | <0.01 | | |

OVS, overlap syndrome

## Symptoms and clinical presentation of OVS compared to OSA

Usual symptoms related to sleep-disordered breathing were generally less reported in the OVS population compared to classical OSA, whereas the severity of AHI was in the same range (Table 3). A lower proportion of OVS patients complained of snoring, morning headaches and excessive daytime sleepiness as measured by the Epworth Sleepiness Scale (median score: 9 [IQR 6 to 13] versus 10 [IQR 6 to 13], respectively; P <0.02). Similarly, a lower proportion of OVS patients (35.6% [95% CI 33.6 to 37.7]) experienced sleepiness while driving compared to OSA patients (39.4% [95% CI 38.6 to 40.3]; P <0.01). There were no differences between groups regarding dyspnea and fatigue as measured by dedicated scales. In contrast, 63.5% (95% CI 61.4 to 65.6) of the OVS population experienced nocturia compared to 58.0% (95% CI 57.2 to 58.8) of the OSA population; P <0.01.

## Cardiovascular and metabolic comorbidities

Both patients with OVS and those with OSA experienced a high burden of prevalent cardiovascular and metabolic comorbidities. Compared to OSA patients, OVS patients had a higher prevalence of hyperlipidemia (35.0% versus 29.7%; P<0.01), hypertension (53.4% versus 43.8%; P<0.01), stroke (4.3% versus 2.8%; P<0.01), coronary artery diseases/myocardial infarction (13.4% versus 7.4%; P<0.01), peripheral arteriopathy (6.0% versus 1.9%; P<0.01), and heart failure (4.4% versus 2.2%; P<0.01). Table 4 shows the associations by multivariable analyses between cardiometabolic comorbidities in OVS patients compared to OSA patients as a reference. After adjustment for age, gender, BMI, exposure to cigarette smoke and AHI, peripheral arterial disease remained associated with OVS patients (odds ratio 1.88; 95% CI 1.50 to 2.36; P < .01). Table 5 and Fig 2 provide a numerical and graphical illustration of how

**Table 3. Symptoms and clinical presentation of OVS compared to OSA.**

| Variable | Items | ALL n = 16,466 | OSA n = 14,368 | Overlap n = 2098 | P |
|---|---|---|---|---|---|
| Depression | | 2,217 (13.5) | 1,966 (13.7) | 251 (12.1) | 0.04 |
| Snoring | | 15,296 (92.9) | 13,393 (93.2) | 1,903 (90.7) | <0.01 |
| Sleepiness at the wheel | | 6,411 (38.9) | 5,665 (39.4) | 746 (35.6) | <0.01 |
| Near-miss accidents | | 449 (2.8) | 398 (2.8) | 51 (2.5) | 0.39 |
| Epworth Sleepiness Scale | | 10 [6; 13] | 10 [6; 13] | 9 [6; 13] | 0.02 |
| Pichot's Fatigue Scale | | 12 [6; 19] | 12 [6; 19] | 12 [6; 19] | 0.08 |
| Morning tiredness | | 11,931 (72.5) | 10,426 (72.6) | 1,505 (71.7) | 0.43 |
| Morning headaches | | 5,933 (36) | 5,294 (36.8) | 639 (30.5) | <0.01 |
| Exertional dyspnea | | 8,581 (52.1) | 7,491 (52.1) | 1,090 (52.0) | 0.88 |
| Nocturia | | 9,670 (58.7) | 8,337 (58.0) | 1,333 (63.5) | <0.01 |
| Number of nocturnal urinations | 1 | 2,615 (33.6) | 2,334 (34.4) | 281 (28.5) | <0.01 |
| | 2 | 2,781 (35.8) | 2,421 (35.7) | 360 (36.5) | |
| | ≥3 | 2,381 (30.6) | 2,035 (30.0) | 346 (35.1) | |
| Restless leg syndrome | | 2,539 (15.6) | 2,194 (15.4) | 345 (16.6) | 0.15 |

OVS = overlap syndrome, OSA = obstructive sleep apnea.

OSA and COPD interact to impact on cardiovascular health. The probability of having hypertension was higher in severe OSA (AHI ≥30) compared to moderate OSA (AHI <30), but was not impacted by COPD. In moderate and severe OSA, COPD was associated with a higher probability of having coronary heart disease, heart failure and, particularly, peripheral arteriopathy in a dose-response relationship. Finally, irrespective of its severity, COPD did not increase the prevalence of stroke in OSA ($P = 0.89$).

A subgroup analysis in patients diagnosed by respiratory polygraphy or polysomnography yielded to similar results.

## Discussion

In this large real-life population of more than 16,000 patients with moderate-to-severe OSA with available spirometry data, the prevalence of OVS was 13%. The burden of symptoms related to sleep-disordered breathing was significantly reduced in OVS compared to OSA alone, although the AHI scores were in the same range and nocturnal hypoxemia was more pronounced. A significant percentage of OSA patients with hypercapnia were obese representing an obesity hypoventilation population included in the general OSA population. After careful consideration of confounders, we suggest that COPD, in addition to moderate or severe OSA, does not have the same impact on the different facets of cardiovascular disease. Prevalence of coronary heart disease, heart failure and peripheral arteriopathy were increased, but we did not observe any effect of COPD on prevalent hypertension and stroke in OSA patients.

We found a prevalence of OVS of 13%, which is in line with most recent epidemiological studies of patients with both OSA and COPD. Approximately 1 out of 10 patients with one disorder will have the other disorder by chance alone [31, 32]. This prevalence most probably reflects the fact that pulmonary function tests were offered in a rather large and unselected OSA population. Specific environmental exposures were associated with OVS compared to the OSA population. Tobacco and alcohol consumption were expected to be higher in COPD and might also predispose to the occurrence or worsening of OSA in subjects by increasing upper airway inflammation and altering protective upper airway reflexes [33]. Reduced physical activity and further physical deconditioning perpetuates the COPD vicious circle, but also

**Table 4. Effect of OVS on associated comorbidities in a multivariable logistic regression model.**

| Variable | OR [95% CI] | *P* value |
|---|---|---|
| Hyperlipidemia | 0.91 [0.82; 1.01] | 0.0722 |
| Hypertension | 0.98 [0.89; 1.09] | 0.7502 |
| Stroke | 1.07 [0.84; 1.36] | 0.5925 |
| Coronary artery disease and myocardial infarction | 1.1 [0.95; 1.28] | 0.2115 |
| Peripheral arteriopathy | 1.91 [1.52; 2.4] | <0.0001 |
| Heart failure | 1.27 [0.99; 1.62] | 0.0644 |

*Odds ratio (OR) is the effect of overlap syndrome (OVS) on different cardiovascular comorbidities after adjustment for age, gender, body mass index, exposure to cigarette smoke, the apnea-hypopnea index and the type of sleep study (type III or type I).

favors leg edema and rostral night fluid shift, a condition associated with OSA, even in non-obese subjects [34]. In a previous study, we have demonstrated that OSA phenotypes are specifically influenced by the above-mentioned environmental factors [9]. Thus, taking into consideration the patient's ecosystem will provide a sound basis for personalized medicine and a multimodal management of OSA, including lifestyle interventions and an exercise programme to complement continuous positive airway pressure (CPAP) treatment [35].

Whereas OSA severity assessed by AHI was similar in both groups, OVS patients exhibited less symptoms related to sleep-disordered breathing and reported less excessive daytime sleepiness compared to the OSA alone population. It can be hypothesized that a more sedentary lifestyle in OVS patients is associated with less self-perceived excessive daytime sleepiness. This difference regarding sleepiness is statistically significant but under the Epworth Sleepiness Scale minimum Clinically Important Difference of -2 or -3 [36,37]. Conversely, nocturia, which disrupts sleep continuity and impacts on daytime sleepiness, was more frequent in the OVS group. Nocturia is a ubiquitous phenomenon associated with several chronic disease and reflects an increased burden of sympathetic activity [38]. It is also an independent risk factor for cardiovascular disease and mortality [39]. Thus, the higher prevalence of nocturia and increased number of voids in our OVS population might reflect an increase in sympathetic activity and raised cardiovascular risk. Fatigue and dyspnea complaints were similar between groups. This can probably be explained by the fact that the two chronic diseases are associated with these complaints, but potentially for different reasons and by different mechanisms. Our data are in line with previous small observational studies [11, 20] showing that classical OSA symptoms are less sensitive to diagnose OSA in COPD. Differences in the burden of symptoms between groups were statistically significant, but too small to suspect or to rule out COPD in single individuals. Other diagnostic methods incorporating pulmonary function tests [10, 11] or high-resolution computed tomography imaging [12] should be designed and validated to identify OSA in COPD populations.

While BMI was in the same range in both groups, nocturnal hypoxia was of greater severity and nearly 20% of OVS patients presented with increased daytime $PaCO_2$. A recent study evidenced that a hypoxic burden in OSA strongly predicts cardiovascular disease-related mortality [40]. Diurnal and nocturnal hypoxemia of COPD is also associated with elevated sympathetic nerve activity, which is probably linked to an increased cardiovascular risk [40]. This is of particular importance when considering the cardiovascular risk in OVS patients [22]. Previous observational studies have suggested a synergistic impact of COPD and OSA on cardiovascular disease [21, 22, 41, 42, 43]. Although these studies generally reported a higher prevalence of cardiovascular disease in OVS, they were flawed by small sample sizes and poor

**Table 5. Effect of COPD and OSA on associated comorbidities in multivariable logistic regression.**

| Variable | | | OR [95% CI] | P value |
|---|---|---|---|---|
| Hypertension | AHI<30 | COPD stage 1 | 0.82 [0.64; 1.06] | <0.0001 |
| | | COPD stage 2 | 1.33 [1.04; 1.69] | |
| | | COPD stages 3 & 4 | 0.81 [0.52; 1.26] | |
| | AHI≥30 | no COPD | 1.42 [1.32; 1.53] | |
| | | COPD stage 1 | 1.3 [1.09; 1.56] | |
| | | COPD stage 2 | 1.7 [1.43; 2.02] | |
| | | COPD stages 3 & 4 | 1.61 [1.16; 2.23] | |
| Peripheral arteriopathy | AHI<30 | COPD stage 1 | 1.18 [0.53; 2.61] | <0.0001 |
| | | COPD stage 2 | 2.91 [1.79; 4.74] | |
| | | COPD stages 3 & 4 | 2.92 [1.27; 6.7] | |
| | AHI≥30 | no COPD | 1.31 [0.99; 1.72] | |
| | | COPD stage 1 | 2.24 [1.44; 3.5] | |
| | | COPD stage 2 | 2.4 [1.6; 3.59] | |
| | | COPD stages 3 & 4 | 2.42 [1.27; 4.63] | |
| Heart failure | AHI<30 | COPD stage 1 | 1.06 [0.53 2.14] | 0.0005 |
| | | COPD stage 2 | 1.58 [0.93; 2.69] | |
| | | COPD stages 3 & 4 | 2.48 [1.14; 5.42] | |
| | AHI≥30 | no COPD | 0.92 [0.72; 1.17] | |
| | | COPD stage 1 | 0.43 [0.22; 0.87] | |
| | | COPD stage 2 | 1.4 [0.93; 2.1] | |
| | | COPD stages 3 & 4 | 2.16 [1.16; 3.99] | |
| Coronary artery disease and myocardial infarction | AHI<30 | COPD stage 1 | 0.76 [0.48; 1.23] | 0.0156 |
| | | COPD stage 2 | 1.02 [0.7; 1.49] | |
| | | COPD stages 3 & 4 | 1.31 [0.69; 2.46] | |
| | AHI≥30 | no COPD | 1.15 [1; 1.33] | |
| | | COPD stage 1 | 1.13 [0.85; 1.5] | |
| | | COPD stage 2 | 1.41 [1.09; 1.81] | |
| | | COPD stages 3 & 4 | 1.88 [1.24; 2.86] | |
| Stroke | AHI<30 | COPD stage 1 | 1.36 [0.75; 2.45] | 0.8969 |
| | | COPD stage 2 | 1.26 [0.72; 2.2] | |
| | | COPD stages 3 & 4 | 1.13 [0.4; 3.17] | |
| | AHI≥30 | no COPD | 1.1 [0.88; 1.36] | |
| | | COPD stage 1 | 0.91 [0.56; 1.48] | |
| | | COPD stage 2 | 1.22 [0.81; 1.84] | |
| | | COPD stages 3 & 4 | 0.91 [0.39; 2.12] | |

*The odds ratio (OR) assesses the effect of COPD and OSA severity on comorbidities by using a logistic regression adjusted on age, gender, exposure to cigarette smoke and the type of sleep study (type III or type I). Patients with moderate sleep apnea and no COPD serve as reference.

AHI, apnea-hypopnea index

characterization of the respective contribution of OSA and COPD. A recent review [44] stated that "careful consideration of the severity of both OSA and COPD is required when determining the significance of OVS as a clinically relevant entity" associated with hard outcomes. This is the case in our current dataset demonstrating that decreasing lung function, in addition to moderate-to-severe OSA, had different effects on different cardiovascular traits. However, it is difficult to speculate why an increasing severity of COPD increased the prevalence of coronary heart disease, heart failure and peripheral arteriopathy in a dose-response manner, while

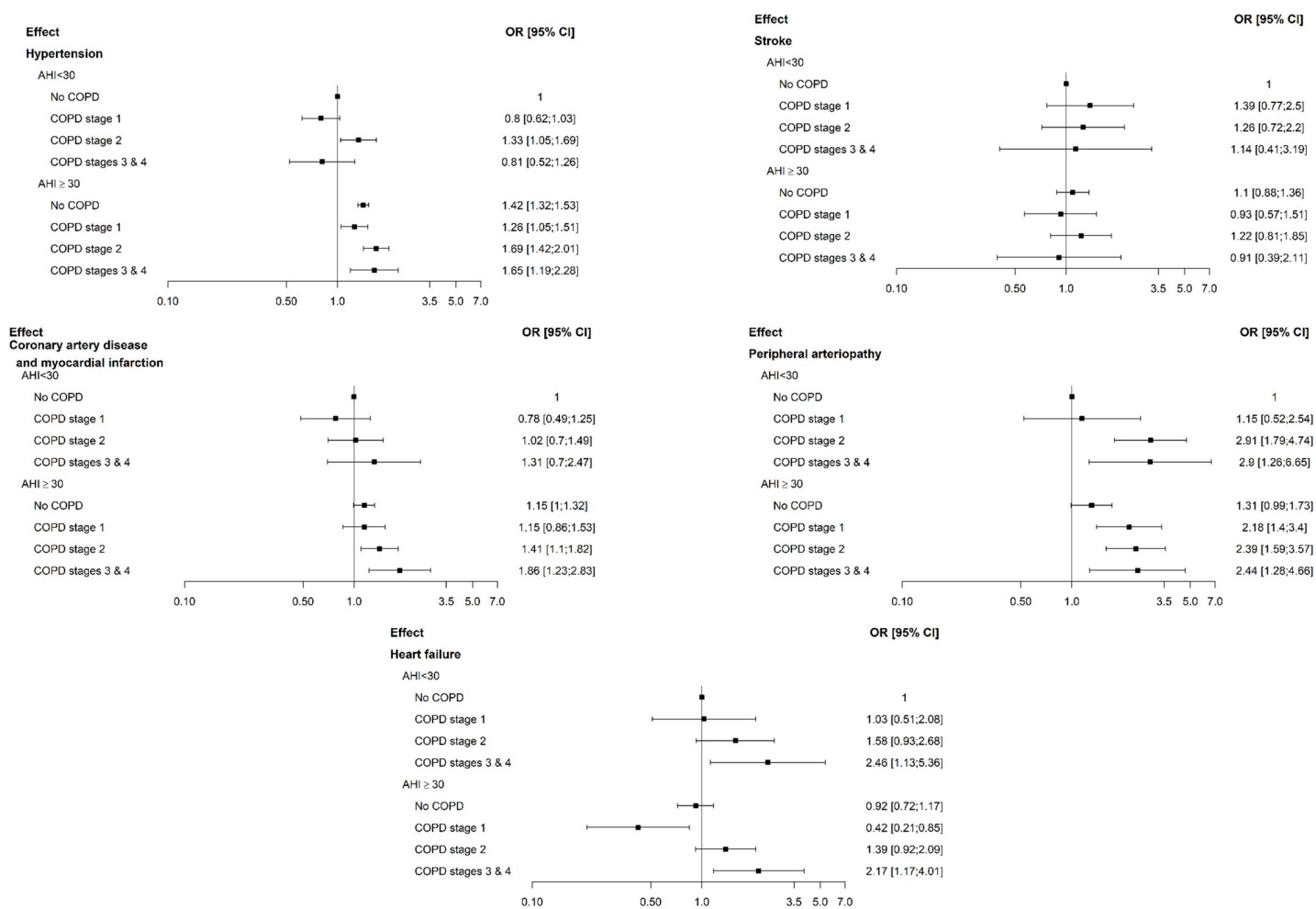

**Fig 2. Graphic illustration of how OSA and COPD interact to impact on cardiovascular health.** The probability of having hypertension was higher in severe OSA (AHI ≥30) compared to non-severe OSA (AHI <30) and was not impacted by COPD. In moderate-to-severe OSA, COPD was associated with a higher probability of having coronary heart disease, heart failure, particularly peripheral arteriopathy, in a dose-response relationship. COPD did not increase the prevalence of stroke in OSA, irrespective of its severity. Odd ratios are adjusted on age, gender and smoking status. Numerical data are provided in Table 4.

prevalent hypertension and stroke seemed not to be affected. Available evidence only suggests that sympathetic activation associated with OSA is more important in systemic hypertension [45], whereas systemic inflammation associated with COPD may play a greater role in the pathogenesis of atherosclerosis and coronary heart disease [46].

Our study has several strengths and limitations. We report here the largest OVS population to date. Both OSA and COPD severity were objectively assessed and allowed subgroup analyses. In such a well-maintained registry, a careful analysis of confounders allowed reporting an association between COPD and cardiovascular health in OSA patients [47] but causality remains to be firmly established by incident data and randomized controlled trials as our study design only allows generating hypothesis. A selection bias is possible since pulmonary function tests were conducted only in a large subset of patients, but not in the whole population referred for a suspicion of OSA [30]. However, a prevalence of OVS around 13% and an analysis of the clinical characteristics of 16,466 OSA patients with available spirometry data compared to 30,320 OSA patients without spirometry data did not bring to light major differences between populations to support a significant selection bias. Regarding dataset, we

acknowledge two more limitations: (i) the majority of centers were using a 3% desaturation and a micro arousal ending the event for hypopneas scoring during polysomnography and reduced flow associated with a 3% desaturation for hypopnea scoring during type III sleep recording. Owing to the size of the registry it was impossible to use a centralized scoring for all sleep studies but we did not observe major differences in AHI between polysomnography and type III sleep recording (not reported); (ii) no detailed information on the type of stroke was available in this large registry.

## Conclusions

We found evidence that comorbid COPD is minimally symptomatic for sleep-disordered breathing but exhibit higher odds for coronary heart disease, heart failure and peripheral arteriopathy. Owing to this specific clinical presentation, new diagnostic tools should be developed to target this particular phenotype to CPAP treatment in order to improve cardiovascular outcome. Randomized controlled trials comparing clinical outcomes among OVS patients with OSA treated with CPAP versus OVS patients with untreated OSA are highly desirable.

## Supporting information

**S1 Table. Clinical characteristics and cardiovascular comorbidities of patients without and with available spirometry data.**
(DOCX)

## Author Contributions

**Conceptualization:** Dan Adler, Marie Joyeux-Faure, Paola Marina Soccal, Jean Paul Janssens, Marc Sapène, Renaud Tamisier, Jean-Louis Pépin.

**Data curation:** Sébastien Bailly, Meriem Benmerad, Marc Sapène, Yves Grillet, Bruno Stach, Renaud Tamisier, Jean-Louis Pépin.

**Formal analysis:** Dan Adler, Sébastien Bailly, Meriem Benmerad.

**Funding acquisition:** Dan Adler, Marie Joyeux-Faure, Yves Grillet, Bruno Stach, Jean-Louis Pépin.

**Investigation:** Dan Adler, Marie Joyeux-Faure, Marc Sapène, Yves Grillet, Bruno Stach, Renaud Tamisier, Jean-Louis Pépin.

**Methodology:** Dan Adler, Sébastien Bailly, Meriem Benmerad, Ingrid Jullian-Desayes, Jean-Louis Pépin.

**Project administration:** Jean-Louis Pépin.

**Resources:** Marc Sapène, Yves Grillet.

**Supervision:** Marc Sapène, Jean-Louis Pépin.

**Validation:** Ingrid Jullian-Desayes, Paola Marina Soccal, Jean Paul Janssens, Renaud Tamisier, Jean-Louis Pépin.

**Visualization:** Jean Paul Janssens, Renaud Tamisier, Jean-Louis Pépin.

**Writing – original draft:** Dan Adler, Ingrid Jullian-Desayes, Jean-Louis Pépin.

**Writing – review & editing:** Marie Joyeux-Faure, Paola Marina Soccal, Jean Paul Janssens, Renaud Tamisier.

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
