## [Decision Letter · Decision Letter 0]

23 Mar 2020

PONE-D-19-30703

Clinical presentation and cardiovascular outcomes of obstructive sleep apnea-COPDoverlap syndrome

PLOS ONE

Dear Dr Adler,

Thank you for submitting your manuscript to PLOS ONE. After careful consideration, we feel that it has merit but does not fully meet PLOS ONE’s publication criteria as it currently stands. Therefore, we invite you to submit a revised version of the manuscript that addresses the points raised during the review process.

First of all, receive my apologies for the delayed process. I was the last handling editor and did my best to terminate the process in a proper time schedule. Both reviewers found your work interesting but simultaneously raised some points mainly related to either methodological issues or/and the way that results were presented. 

We would appreciate receiving your revised manuscript by May 07 2020 11:59PM. To enhance the reproducibility of your results, we recommend that if applicable you deposit your laboratory protocols in protocols.io, where a protocol can be assigned its own identifier (DOI) such that it can be cited independently in the future. For instructions see: http://journals.plos.org/plosone/s/submission-guidelines#loc-laboratory-protocols

We look forward to receiving your revised manuscript.

Kind regards,

Stelios Loukides

Academic Editor

PLOS ONE

Journal Requirements:

"This study was funded by an unrestricted grant from “Appel d’offre cohorte-fondation du souffle” (SRC_2017). SB, RT and JLP are supported by a research grant from the French National Research Agency (ANR-12-TECS-0010) in the framework of the “Investissements d’avenir” programme (ANR-15-IDEX-02) and the “e-health and integrated care” Chair of excellence of the University Grenoble Alpes Foundation. DA is supported by a research grant from the “Ligue Pulmonaire Suisse” to conduct a programme entitled “integrated care of patients surviving acute hypercapnic respiratory failure in the ICU”."

We note that one or more of the authors are employed by a commercial company: Private Practice Sleep and Respiratory Disease Centre.

Reviewers' comments:

Reviewer's Responses to Questions

**Comments to the Author**

1. Is the manuscript technically sound, and do the data support the conclusions?

Reviewer #1: No

Reviewer #2: Yes

2. Has the statistical analysis been performed appropriately and rigorously? 

Reviewer #1: Yes

Reviewer #2: Yes

3. Have the authors made all data underlying the findings in their manuscript fully available?

Reviewer #1: No

Reviewer #2: Yes

4. Is the manuscript presented in an intelligible fashion and written in standard English?

Reviewer #1: Yes

Reviewer #2: Yes

5. Review Comments to the Author

Reviewer #1: This cross-sectional analysis of the French Sleep Apnoea Registry addresses an interesting question, namely how the co-existence of COPD and OSA (Overlap Syndrome) affects symptoms comorbidities compared to OSA alone. The sample size is large, however, the study design introduces a lot of bias that hampers the interpretation of the findings. In addition, statistically significant differences in symptom burden are reported between Overlap Syndrome and COPD alone. However, when looking at the difference and the confidence intervals, there is no clinically relevant difference between the groups. Therefore, the conclusion is not really supported by the data.

The question asked and the data are interesting but the title, the methods (hypothesis, outcomes), the interpretation of the data, the limitations and the conclusions need to be revised thoroughly.

- mean (SD) or median (IQR) AHI, ODI and SpO2 should be reported in the abstract to understand differences in SDB in the two compared populations

- the method section of the abstract should report outcomes of interest and statistical approach

- the difference in ESS between the compared populations is statistically significant but there is no clinically meaningful difference (MCID in ESS 2-3), therefore, the conclusion is not supported by the data

- patients with OSA who are in the French register represent a population referred to a sleep clinic due to suspicion of symptomatic SDB whereas patients with Overlap Syndrome might have undergone a sleep study for a different reason; this introduces a relevant selection bias that hampers the interpretation on differences in sleepiness and associated symptoms

- due to the study design, these results can be used for generating a hypothesis but not more

- the title should be changed to symptoms and comorbidities – delete cardiovascular outcomes); this is a cross-sectional analysis not a longitudinal analysis that looks at how baseline characteristics affect the incidence of cardiovascular events

- it should be discussed which scoring rules have been used during the period analysed (1997-2017); hypopnea definition has changed and this will affect the prevalence of Overlap Syndrome

Reviewer #2: This cross-sectional study evaluates and compares the clinical characteristics of patients diagnosed with OVS and moderate OSA alone, in the largest until now study, retrieved from the French National Sleep Apnea Registry. In addition, emphasis was placed on the impact that COPD may have on cardiovascular outcomes in patients with OSA. COPD diagnosis was based on available spirometric data, while OSA was diagnosed using either PSG or type III cardio-respiratory studies. The percentage of OVS patients out of the total examined OSA population was in accordance with published epidemiological studies on this field (at 13%), whereas OSA related symptoms were less frequently reported in the OVS than the OSA population (apart from nocturnia). Finally, and more importantly, increasing COPD severity, based on GOLD airflow limitation criteria, associated with higher prevalence of CVD, but not with hypertension and stroke after adjustment for significant cofounders.

Major concerns

1. It would be interesting to distinguish between patients diagnosed with PSG or type III studies and to perform the same analyses. The findings of this proposed subgroup analysis may explain possible differences regarding the impact of COPD on prevalent hypertension as well as other important findings such as symptoms related to OSA between investigated groups.

2. The authors should consider the fact that hypercapnic OSA patients (6.7% of the patients in this study) may have obesity hypoventilation syndrome, especially since the reported median BMI is 31.1 kg/m2. FVC values could also be helpful in this. I would suggest to add this this, at least as a limitation because those patients may not represent a “pure” OSA population.

3. It would be also useful to better define COPD, underling the fact that included patients should be ≥40 years and have a history and symptoms compatible with a COPD diagnosis. How the authors explain that 33.7% of the OVS patients were non-smokers?

4. The authors should also add minimum SaO2 values dying sleep in Table 1. They should also remove peripheral arteriopathy from Table1 or to add other comorbidities.

5. In the present study, OVS was not associated with prevalent stroke. Were there any available data regarding the subtypes of stroke in this study (i.e. ischaemic or haemorragic)? Actually, COPD may especially increase the risk of hemorrhagic stroke, while the risk of ischemic stroke in COPD patients could be confounded by other parameters. Similarly, non- significant effect of COPD on prevalent hypertension may be hard to explain. Nevertheless, moderate to severe and very severe COPD are all associated with multimorbidity, making it difficult to find any difference in prevalent hypertension.

Minor concerns

1. The authors state that “Randomized controlled trials comparing clinical outcomes among OVS patients with OSA treated with CPAP versus OVS patients with untreated OSA are highly desirable.”

Although RCTs are the cornerstone of medical evidence and explain causality, they are hard to be implemented in daily practice due to ethics concerns regarding treatment of OSA. Thus, real life studies, as the present one, could offer important evidence on the field of OVS.

2. Discussion section: the authors state that “Sustained hypoxemia may also adequately assess the degree of sympathetic activity.” They should provide more clarifications and references.

6. PLOS authors have the option to publish the peer review history of their article (what does this mean?). If published, this will include your full peer review and any attached files.

Reviewer #1: No

Reviewer #2: Yes: Paschalis Steiropoulos

---

## [Decision Letter · Decision Letter 1]

15 Jun 2020

Clinical presentation and comorbidities of obstructive sleep apnea- COPD overlap syndrome

PONE-D-19-30703R1

Dear Dr. Adler,

We’re pleased to inform you that your manuscript has been judged scientifically suitable for publication and will be formally accepted for publication once it meets all outstanding technical requirements.

Kind regards,

Stelios Loukides

Academic Editor

PLOS ONE

Additional Editor Comments (optional):

Reviewers' comments:

Reviewer's Responses to Questions

**Comments to the Author**

1. If the authors have adequately addressed your comments raised in a previous round of review and you feel that this manuscript is now acceptable for publication, you may indicate that here to bypass the “Comments to the Author” section, enter your conflict of interest statement in the “Confidential to Editor” section, and submit your "Accept" recommendation.

Reviewer #1: All comments have been addressed

Reviewer #2: All comments have been addressed

2. Is the manuscript technically sound, and do the data support the conclusions?

Reviewer #1: Yes

Reviewer #2: Yes

3. Has the statistical analysis been performed appropriately and rigorously? 

Reviewer #1: I Don't Know

Reviewer #2: Yes

4. Have the authors made all data underlying the findings in their manuscript fully available?

Reviewer #1: Yes

Reviewer #2: Yes

5. Is the manuscript presented in an intelligible fashion and written in standard English?

Reviewer #1: Yes

Reviewer #2: Yes

6. Review Comments to the Author

Reviewer #1: Reviewer‘s comments have been addressed. The new version is fine and these registry data are of interest and should be made available to our research community

Reviewer #2: (No Response)

7. PLOS authors have the option to publish the peer review history of their article (what does this mean?). If published, this will include your full peer review and any attached files.

Reviewer #1: Yes: Esther Irene Schwarz

Reviewer #2: Yes: Paschalis Steiropoulos

---

## [Editor Report · Acceptance letter]

18 Jun 2020

PONE-D-19-30703R1 

Clinical presentation and comorbidities of obstructive sleep apnea-COPD overlap syndrome 

Dear Dr. Adler:

I'm pleased to inform you that your manuscript has been deemed suitable for publication in PLOS ONE. Congratulations! Your manuscript is now with our production department. 

Kind regards, 

on behalf of

Dr Stelios Loukides 

Academic Editor

PLOS ONE